# Deep Gamblers:
# Learning to Abstain with Portfolio Theory

**Liu Ziyin**[†]**, Zhikang T. Wang**[†]**, Paul Pu Liang**[♭]**,**
**Ruslan Salakhutdinov**[♭]**, Louis-Philippe Morency**[◊]**, Masahito Ueda**[†]
[†]Institute for Physics of Intelligence & Department of Physics, University of Tokyo
[♭]Machine Learning Department, Carnegie Mellon University
[◊]Language Technologies Institute, Carnegie Mellon University
{zliu,wang}@cat.phys.s.u-tokyo.ac.jp ueda@phys.s.u-tokyo.ac.jp
{pliang,rsalakhu,morency}@cs.cmu.edu

## Abstract

We deal with the *selective classification* problem (supervised-learning problem with a rejection option), where we want to achieve the best performance at a certain level of coverage of the data. We transform the original $m$-class classification problem to $(m+1)$-class where the $(m+1)$-th class represents the model abstaining from making a prediction due to disconfidence. Inspired by portfolio theory, we propose a loss function for the selective classification problem based on the doubling rate of gambling. Minimizing this loss function corresponds naturally to maximizing the return of a *horse race*, where a player aims to balance between betting on an outcome (making a prediction) when confident and reserving one's winnings (abstaining) when not confident. This loss function allows us to train neural networks and characterize the disconfidence of prediction in an end-to-end fashion. In comparison with previous methods, our method requires almost no modification to the model inference algorithm or model architecture. Experiments show that our method can identify uncertainty in data points, and achieves strong results on SVHN and CIFAR10 at various coverages of the data.

## 1 Introduction

With deep learning's unprecedented success in fields such as image classification [21, 18, 24], language understanding [9, 35, 42, 32], and multimodal learning [26, 33], researchers have now begun to apply deep learning to facilitate scientific discovery in fields such as physics [2], biology [38], chemistry [16], and healthcare [20]. However, one important challenge for applications of deep learning to these natural science problems comes from the requirement of assessing the confidence level in prediction. Characterizing confidence and uncertainty of model predictions is now an active area of research [12], and being able to assess prediction confidence allows us to handpick difficult cases and treat them separately for better performance [13] (e.g., by passing to a human expert). Moreover, knowing uncertainty is important for fundamental machine learning research [19]; for example, many reinforcement learning algorithms (such as Thompson sampling [40]) requires estimating uncertainty of the distribution [39].

However, there has not been any well-established, effective and efficient method to assess prediction uncertainty of deep learning models. We believe that there are four desiderata for any framework to assess deep learning model uncertainty. Firstly, they must be *simply end-to-end trainable*, because end-to-end trainability is important for accessibility to the method. Secondly, it should require *no heavy sampling procedure* because sampling a model or prediction (as in Bayesian methods) hundreds of times is computationally heavy. Thirdly, it *should not require retraining* when different levels of uncertainty are required because many tasks such as ImageNet [8] and 1 Billion Word [4] require

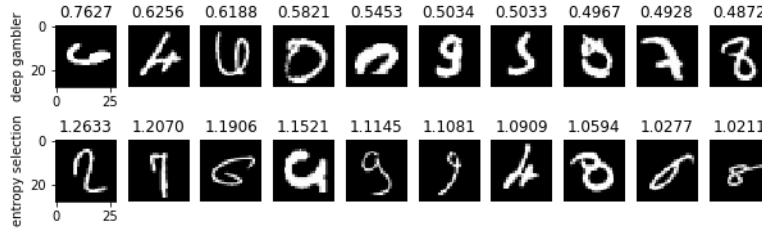

Figure 1: Top-10 rejected images in the MNIST testing set found by two methods. The number above image is the predicted uncertainty score (ours) or the entropy of the prediction (baseline). For the top-2 images, our method chooses images that are hard to recognize, while that of the baseline can be identified unambiguously by human.

weeks of training, which is too expensive. Lastly, it *should not require any modification to existing model architectures* so that we can achieve better flexibility and minimal engineering effort. However, most of the methods that currently exist do not meet some of the above criteria. For example, existing Bayesian approaches, in which priors of model parameters are defined and the posteriors are estimated using the Bayes theorem [29, 13, 34], usually rely heavily on sampling to estimate the posterior distribution [13, 34] or modifying the network architecture via the reparametrization trick [22, 3]. These methods therefore incur computational costs which slow down the training process. This argument also applied to ensembling methods [25]. Selective classification methods offer an alternative approach [5], which either require modifying model objectives and retraining the model [14, 15]. See Table 1 for a summary of the existing methods, and the problems with these methods are discussed in Section 3. In this paper, we follow the selective classification framework (see Section 2), and focus on a setting where we only have a single classifier augmented with a rejection option[1]. Inspired by portfolio theory in mathematical finance [30], we propose a loss function for the selective classification problem that is easy to optimize and requires almost no modification to existing architectures.

## 2 The Selective Prediction Problem

In this work, we consider a selective prediction problem setting [10]. Let $X$ be the feature space and $Y$ the label space. For example, $X$ could be the distribution of images, and $Y$ would be the distribution of the class labels, and our goal is to learn the conditional distribution $P(Y|X)$, and a *prediction model* parametrized by weight $\mathbf{w}$ is a function $f_{\mathbf{w}} : X \to Y$. The *risk* of the task w.r.t to a loss function $\ell(\cdot)$ is $\mathbb{E}_{P(X,Y)}[\ell(f(x), y)]$, given a dataset with size $N$ $\{(x_i, y_i)\}_{i=1}^{N}$ where all $(x_i, y_i)$ are independent draws from $X \times Y$. A prediction model augmented with a rejection option is a pair of functions $(f, g)$ such that $g_h : X \to \mathbb{R}$ is a selection function which can be interpreted as a binary qualifier for $f$ as follow:

$$(f, g)(x) := \begin{cases} f(x), & \text{if } g_h(x) \geq h \\ \text{ABSTAIN}, & \text{otherwise} \end{cases} \tag{1}$$

i.e., the model abstains from making a prediction when the selection function $g(x)$ falls below a predetermined threshold $h$. We call $g(x)$ the uncertainty score of $x$; different methods tend to use different $g(x)$. The *covered dataset* is defined to be $\{x : g_h(x) \geq h\}$, and the *coverage* is the ratio of the size of covered dataset to the size of the original dataset. Clearly, one may trade-off coverage for lower risk, and this is the motivation behind rejection option methods.

## 3 Related Work

**Abstention Mechanisms**. Here we summarize the existing methods to perform abstention, and these are the methods we will compare with in this paper. For a summary of the features of these methods, see table 1. **Entropy Selection (ES)**: This is the simplest way to output an uncertainty score for a prediction, we compare with this in the qualitative experiments. It simply takes the entropy of the predicted probability as the uncertainty score. **Softmax-Response (SR, [14])**: This is a simple yet theoretically-guaranteed strong baseline proposed in [14]. It regards the maximum predicted probability as the confidence score; it differs from our work in that it does not involve *training* an

| | Ours | SR [14] | BD [13] | SN [15] |
|---|---|---|---|---|
| Simple end-to-end training | ✓ | ✓ | ✓ | ✗ |
| No sampling process required | ✓ | ✓ | ✗ | ✓ |
| No retraining needed for different coverage | ✓ | ✓ | ✓ | ✗ |
| No modification to model architecture | ✓ | ✓ | ✓ | ✗ |

Table 1: Summary of features of different methods for selective prediction. Our method is end-to-end trainable and does not require sampling, retraining, or architecture modification.

abstention mechanism. **Bayes Dropout (BD, [13])**: This is a SOTA Bayesian method that offer a way to reject uncertain images [13]. One problem with with method is that one often needs about extensive sampling to obtain an accurate estimation of the uncertainty. **SelectiveNet (SN, [15])**: This is a very recent work that also trains a network to predict its uncertainty, and is the current SOTA method of the selective prediction problem. The loss function of this method requires interior point method to optimize and depends on the target coverage one wants to achieve.

**Portfolio Theory and Gambling** The Modern Portfolio Theory (MPT) is a modern method in investment for assembling a portfolio of assets that maximizes expected return while minimizing the risk [30]. The generalized portfolio theory is a constrained minimization problem in which we sought for maximum expected return with a variance constraint. In this work, however, we explore a very limited form of portfolio theory that can be seen as a *horse race*, as a proof of concept for bridging uncertainty in deep learning and portfolio theory. In this work, we focus on the classification problem, and we believe that regression problems can similarly be reformulated as a general portfolio problem, and we leave this line of research to the future. The connection between portfolio theory, gambling and information theory is studied in [6, 7]. Some of the theoretical arguments presented in this work are based on arguments given in [7].

## 4 Learning to Abstain with Portfolio Theory

The intuition behind the method is that a deep learning model learning to abstain from prediction indeed mimicks a gambler learning to reserve betting in a game. Indeed, we show that if we have a $m$-class classification problem, we can instead perform a $m + 1$ class classification which predicts the probabilities of the $m$ classes and use the $(m + 1)$-th class as an additional rejection score. This method is similar to [14, 15], and the difference lies in how we learn such a model. We use ideas from portfolio theory which says that if we have some budget, we should split them between how much we would like to bet, and how much to save. In the following sections, we first provide a gentle introduction to portfolio theory which will provide the mathematical foundations of our method. We then describe how to adapt portfolio theory for classification problems in machine learning and derive our adapted loss function that trains a model to predict a rejection score. We finally prove some theoretical properties of our method to show that a classification problem can indeed be seen as a gambling problem, and thus avoiding a bet in gambling can indeed been interpreted as giving a rejection score.

### 4.1 A Short Introduction to General Portfolio Theory

To keep the terminology clear, we give a chart of the terms from portfolio theory and their corresponding concepts in deep learning in Table 2. The rows in the dictionary show the correspondences we are going to make in this section. In short, portfolio theory tells us what is the best way to invest in a stock market. A stock market with $m$ stocks is a vector of positive real numbers $\mathbf{X} = (X_1, ..., X_m)$, and we define the *price relaive* $X_i$ as the ratio of the price of the stock $i$ at the end of the day to the price at the beginning

| Portfolio Theory | Deep Learning |
|---|---|
| Portfolio | Prediction |
| Doubling Rate | negative NLL loss |
| Stock/Horse | input data point |
| Stock Market Outcome | Target Label |
| Horse Race Outcome | Target Label |
| Reservation in Gamble | Abstention |

Table 2: Portfolio Theory - Deep Learning Dictionary.

of the day. For example, $X_i = 0.95$ means that the price of the stock is $0.95$ times its price at the beginning of the day. We formulate the price vector as a vector of random variables drawn from a joint distribution $\mathbf{X} \sim P(\mathbf{X})$. A *portfolio* refers to our investment in this stock market, and can be modeled as a discrete distribution $\mathbf{b} = (b_1, ..., b_m)$ where $b_i \geq 0$ and $\sum_i b_i = 1$, and $\mathbf{b}$ is our distributing of wealth to $\mathbf{X}$. In this formulation, the *wealth relative* at the end of the day is $S = \mathbf{b}^T \mathbf{X} = \sum_i b_i X_i$; this tell us the ratio of our wealth at the end of the day to our wealth at the beginning of the day.

**Definition 1.** *The doubling rate of a stock market portfolio* **b** *with respect to a stock distribution* $P(\mathbf{X})$ *is*

$$W(\mathbf{b}, P) = \int \log_2\left(\mathbf{b}^T\mathbf{x}\right) dP(\mathbf{x}).$$

This tells us the speed at which our wealth increases, and we want to maximize $W$. Now we consider a simplified version of portfolio theory called the "horse race".

## 4.2 Horse Race

Different from a stock market, a horse race has an exclusive outcome (only one horse wins, and it's either win or loss) $\mathbf{x}(j) = (0, ..., 0, 1, 0, ..., 0)$, which is a one-hot vector on the $j$-th entry. In a horse race, we want to bet on $m$ horses, and the $i$-th horse wins with probability $p_i$, and the payoff is $o_i$ for betting 1 dollar on horse $i$ if $i$ wins, and the payoff is 0 otherwise. Now the gambler can choose to distribute his wealth over the $m$ horses, according to **o** and **p**, and let **b** denote such distribution; this corresponds to choosing a portfolio. Again, we require that $b_i \geq 0$, and $\sum_i b_i = 1$. The *wealth relative* of the gambler at the end of the game will be $S(\mathbf{x}(j)) = b_j o_j$ when the horse $j$ wins. After $n$ many races, our wealth relative would be:

$$S_n = \prod_{i=1}^n S(\mathbf{x}_i). \tag{2}$$

Notice that our relative wealth after $n$ races does not depend on the order of the occurrence of the result of each race (and this will justify our treatment of a batch of samples as races). We can define the doubling rate by changing the integral to a sum:

**Definition 2.** *The doubling rate of a horse race is*

$$W(\mathbf{b}, \mathbf{p}) = \mathbb{E}\log_2(S) = \sum_{i=1}^m p_i \log_2(b_i o_i).$$

As before, we want to maximize the doubling rate. Notice that if we take $o_i = 1$ and $b_i$ be the post-softmax output of our model, then $W$ is equivalent to the commonly used cross-entropy loss in classification. However, a horse race can be more general because the gambler can choose to bet only with part of his money and reserve the rest to minimize risk. This means that, in a horse race with reservation, we can bet on $m + 1$ categories where the $m + 1$-th category denotes reservation with payoff 1. Now the wealth relative after a race becomes $S(\mathbf{x}_j) = b_j o_j + b_{m+1}$ and our objective becomes $\max_{\mathbf{b}} W(\mathbf{b}, \mathbf{p})$, where

$$\max W(\mathbf{b}, \mathbf{p}) = \sum_{i=1}^m p_i \log(b_i o_i + b_{m+1}). \tag{3}$$

This is the *gambler's loss*.

## 4.3 Classification as a Horse Race

An $m$-class classification task can be seen as finding a function $f : \mathbb{R}^n \to \mathbb{R}^m$, where $n$ is the input dimension and $m$ is the number of classes. For an output $f(x)$, we assume that it is normalized, and we treat the output of $f(\cdot)$ as the probability of input $x$ being labeled in class $j$:

$$\Pr(j|x) = f(x)_j \tag{4}$$

Now, let us parametrize the function $f$ as a neural network with parameter **w**, whose output is a distribution over the class labels. We want to maximize the log probability of the true label $j$:

$$\max \mathbb{E}[\log p(j|x)] = \max_{\mathbf{w}} \mathbb{E}[\log f_{\mathbf{w}}(x)_j] \tag{5}$$

For a $m$-class classification task, we transform it to a horse race with reservation by adding a $m+1$-th class, which stands for reservation. The objective function for a mini-batch of size $B$, and for constant $o$ over all categories is then (cf. Equation 3)

$$\max_f W(\mathbf{b}(f), \mathbf{p}) = \max_{\mathbf{w}} \sum_i^B \log\left[f_{\mathbf{w}}(x_i)_{j(i)}o + f_{\mathbf{w}}(x_i)_{m+1}\right]. \tag{6}$$

where $i$ is the index over the batch, and $j(i)$ is the label of the $i$-th data point. As previously remarked, if $o_j = 1$ for all $j$ and $b_{m+1} = 0$, we recover the standard supervised classification task. Therefore $o$

becomes a hyperparameter, and a higher $o$ encourages the network to be confident in inferring, and a low $o$ makes it less confident. In the next section, we will show that the problem is only meaningful for $1 < o < m$. The selection function $g(\cdot)$ is then $f_{\mathbf{w}}(\cdot)_{m+1}$(cf. Equation 1), and prediction on different coverage can be achieved by simply calibrating the threshhold $h$ on a validation set. Also notice that an advantage of our method over the current SOTA method [15] is that our loss function does not depend on the coverage.

## 5   Information Theoretic Analysis

In this section, we analyze our formulation theoretically to explain how our method works. In the first theorem, we show that for a *horce race* without reservation, its optimal solution exists. We then show that, in a setting (gambling with side information) that resembles an actual classification problem, the optimal solution also exists, and it is the same as the optimal solution we expect for a classification problem. The last theorem deals with the possible range of $o$ for a horse race with reservation, followed by a discussion about we should choose the hyperparameter $o$.

In the problem setting, we considered a gambling problem that is probabilistic in nature. It corresponds to a horse race in which, the distribution of winning horses is drawn from a predetermined distribution $P(Y)$ and no other information besides the indices of the horse is given. In this case, we show that the optimal solution should be proportional to $P(Y)$ when no reservation is allowed.

**Theorem 1.** *The optimal doubling rate is given by*

$$W^*(p) = \sum_i p_i \log o_i - H(p). \tag{7}$$

*where $H(p) = -\sum p \log p$ is the entropy of the distribution $p$, and this rate is achieved by proportional gambling $\mathbf{b}^* = p$.*

This result shows the equivalence between a prediction problem and a gambling problem. In fact, trying to minimize the natural log loss for a classification task is the same as trying to maximize the doubling rate in a gambling problem. However, in practice, we are often in a horse race where some information about the horse is known. For example, in the "MNIST" horse race, one sees a picture, and want to guess its category, i.e., one has access to side information. In the next theorem, we show that in a gambling game with side information, the optimal gambling strategy is obtained by a prediction that maximizes the mutual information between the horse (image) and the outcome (label). This is a classical theorem that can be found in [7]. The proofs are given in the appendix.

**Theorem 2.** *Let $W$ denote the doubling rate defined in Def. 2. For a horse race $Y$ to which some side information $X$ is given, the amount of increase $\Delta W$ is*

$$\Delta W = I(X;Y) = \sum_{x,y} p(x,y) \log \frac{p(x,y)}{p(x)p(y)}. \tag{8}$$

This shows that the increase in the doubling rate from knowing $X$ is bounded by the mutual information between the two. This means that the neural network, during training, will have to maximize the mutual information between the prediction and the true label of the sample. This shows that an image classification problem is exactly equal to a horse race with side information. However, the next theorem makes our formulation different from a standard classification task and can be seen as a generalization of it. We show that, when reservation is allowed, the optimal strategy changes with $o$, the return of winning. Especially, for some range of $o$, only trivial solutions to the gambling problem exist. Since the tasks in this work only deals with situations in which $o$ is uniform across categories, we assume $o$ to be uniform for clarity.

**Theorem 3.** *Let $m$ be the number of horses, and let $W$ be defined as in Eq. 3, and let $o_i = o$ for all $i$; then if $o > m$, the optimal betting always have $b_{m+1} = 0$; if $o < 1$, then the optimal betting always have $b_i = 0$ for $i \neq m + 1$.*

This theorem tells us that when the return from betting is too high ($o > m$), then we should always bet, and so the optimal solution is given by Theorem 1; when the return from betting is too low ($o < 1$), then we should always reserve. A more realistic situation should have that $1 < o < m$, which reflects the fact that, while one might expect to gain in a horse race, the organizer of the game takes a cut of the bets. We discuss the effect of varying $o$ in the appendix. In fact, the optimal rejection score of to a given prediction probability of our method can be easily found using Kuhn-Tucker condition

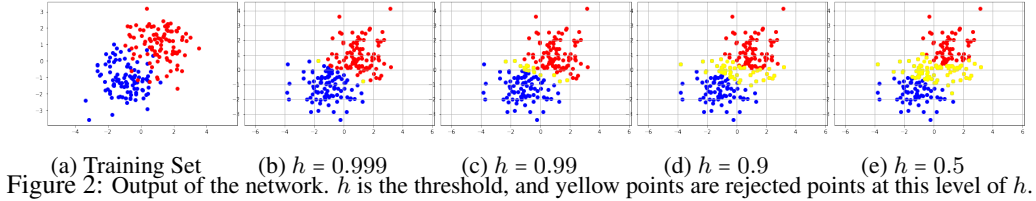

(a) Training Set     (b) $h = 0.999$     (c) $h = 0.99$     (d) $h = 0.9$     (e) $h = 0.5$

Figure 2: Output of the network. $h$ is the threshold, and yellow points are rejected points at this level of $h$.

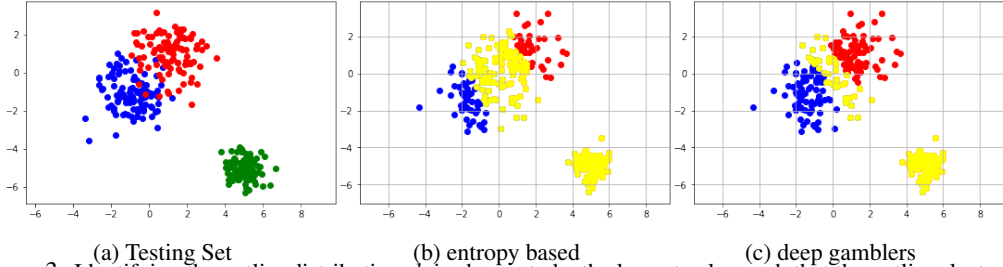

(a) Testing Set     (b) entropy based     (c) deep gamblers

Figure 3: Identifying the outlier distribution. $h$ is chosen to be the largest value such that the outlier cluster is rejected. We see that a network trainied with our method rejects the outlier cluster much earlier than the entropy based method.

without training the network, but we argue that it is not the learned rejection score that is the core of method, but that this loss function allows the trained model to learn a qualitatively different and better hidden representation than the baseline model. See Figure 5 (and appendix).

# 6 Experiments

We begin with some toy experiments to demonstrate that our method can deal with various kinds of uncertain inputs. We then compare the existing methods in selective classification and show that the proposed method is very competitive against the SOTA method. Implementation details are in the appendix.

## 6.1 Synthetic Gaussian Dataset

In this section, we train a network with 2 hidden layers each with 50 neurons and $tanh$ activation. For the training set, we generate 2 overlapping diagonal $2d$-Gaussian distributions and the task is a simple binary classification. Cluster 1 has mean $(1, 1)$ and unit variance, and cluster 2 has mean $(-1, -1)$ with unit variance. The fact that these two clusters are not linearly separable is the first source of uncertainty. A third out-of-distribution cluster exists only in the testing set to study how the model deals with out-of-distribution samples. This distribution has mean $(5, -5)$ and variance $0.5$. This is the second source of uncertainty. Figure 2(a) shows the training set and 3(a) shows the test set.

We gradually decrease the threshold $h$ for the predicted disconfident score, and label the points above the threshold as rejected. These results are visualized in Figure 2 and we observe that the model correctly identifies the border of the two Gaussian distributions as the uncertain region. We also see that, by lowering the threshold, the width of the uncertain region increases. This shows how we might calibrate threshold $h$ to control coverage. Now we study how the model deals with out-of-distribution uncertainty. From Figure 3, we see that the entropy based selection is only able to reject the third cluster when most of data points are excluded, while our method seems to reject the outliers equally well with the boundary points.

## 6.2 Locating the outlier testing images of MNIST

In this section, we show the images that our method finds the most disconfident in MNIST in comparison with the entropy selection method in Figure 1. The model is a simple 4-layer CNN. We find that our method seems to outperform the baseline qualitatively. For example, the two least certain images found by the entropy based method can be labeled by a human unambiguously as a 2 and 7, while the top-2 images found by our method do not look like images of numbers at all. Most figures of this experiment and plots of how the images change across different epochs can be found in the appendix.

## 6.3 Rotating an MNIST image

For illustration, we choose an image of 9 and rotate it up to 180 degrees because a number 9 looks like a distorted 5 when rotated by 90 degrees and looks like a 6 when rotated by 180, which allows

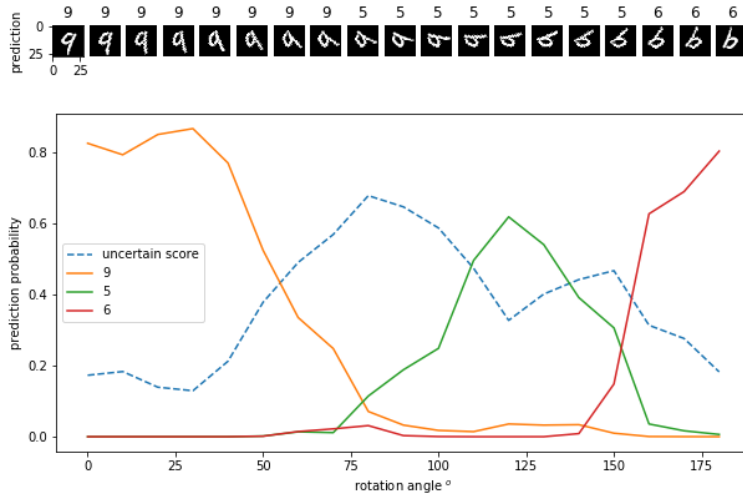

Figure 4: Rotating an image of 9 by 180 degrees. The number above the images are the prediction label of the rotated image.

| Coverage | Ours (Best Single Model) | Ours (Best per coverage) | SR | BD | SN |
|---|---|---|---|---|---|
| 1.00 | $^{o=2.6}3.24 \pm 0.09$ | – | 3.21 | 3.21 | 3.21 |
| 0.95 | $^{o=2.6}\mathbf{1.36 \pm 0.02}$ | $^{o=2.6}\mathbf{1.36 \pm 0.02}$ | 1.39 | 1.40 | 1.40 |
| 0.90 | $^{o=2.6}\mathbf{0.76 \pm 0.05}$ | $^{o=2.6}\mathbf{0.76 \pm 0.05}$ | 0.89 | 0.90 | $0.82 \pm 0.01$ |
| 0.85 | $^{o=2.6}\mathbf{0.57 \pm 0.07}$ | $^{o=3.6}0.66 \pm 0.01$ | 0.70 | 0.71 | $\mathbf{0.60 \pm 0.01}$ |
| 0.80 | $^{o=2.6}\mathbf{0.51 \pm 0.05}$ | $^{o=3.6}0.53 \pm 0.04$ | 0.61 | 0.61 | $0.53 \pm 0.01$ |

Table 3: SVHN. The number is error percentage on the covered dataset; the lower the better. We see that our method achieved competitive results across all coverages. It is the SOTA method at coverage $(0.85, 1.00)$.

us to analyze the behavior of the model clearly. See figure 4. We see that the model assesses its disconfidence as we expected, labeling the image as 9 at the beginning and 6 at the end, and as a 5 with high uncertainty in an intermediate region. We also notice that the uncertainty score has two peaks corresponding to crossing of decision boundaries. This suggests that the model has really learned to assess uncertainty in a subtle and meaningful way (also see Figure 5).

## 6.4 Comparison with Existing Methods

In this section, we compare with the SOTA methods in selective classification. The experiment is performed on SVHN [31] (Table 3), CIFAR10 [23] (Table 4) and Cat vs. Dog (Table 5). We follow exactly the experimental setting in [15] to allow for fair comparison. We use a version of VGG16 that is especially optimized for small datasets [27] with batchnorm and dropout. The baselines we compare against are given in Section 3 and summarized in Table 1. A grid search is done over hyperparameter $o$ with a step size of $0.2$. The best models of ours for a given coverage are chosen using a validation set, which is separated from the test set by a fixed random seed, and the best single model is chosen by using the model that achieves overall best validation accuracy. To report error bars, we estimate its standard deviation using the test errors on neighbouring 3 hyperparameter $o$ values in our grid search (e.g. for $o = 6.5$, the results from $o = 6.3, 6.5, 6.7$ are used to compute the variance).

The results for the baselines are cited from [15], and we show the error bar for the contender models when it overlaps or seem to overlap with our confidence interval. We see that our model achieves SOTA on SVHN on all coverages, in the sense that our model starts at full coverage with a slightly lower accuracy but starts to outperform other contenders starting from $0.95$ coverage, meaning that it learned to identify the hard images better than its contenders. We also perform the experiment on CIFAR-10 and Cat vs. Dog datasets, and we see that our method achieves very strong results. A small problem for the comparison remains since our models have different full coverage performance from other methods, but a closer look suggests that our method performs indeed better when the coverage is in the range $[0.8, 1.0)$ (by comparing the relative improvements). Below $0.8$ coverage, the comparison becomes hard since there are only few images remaining, and methods on different

| Coverage | Ours (Single Best Model) | Ours (Best per Coverage) | SR | BD | SN |
|---|---|---|---|---|---|
| 1.00 | $^{o=2.2}6.12 \pm 0.09$ | – | 6.79 | 6.79 | 6.79 |
| 0.95 | $^{o=2.2}\mathbf{3.49 \pm 0.15}$ | $^{o=6.0}3.76 \pm 0.12$ | 4.55 | 4.58 | 4.16 |
| 0.90 | $^{o=2.2}\mathbf{2.19 \pm 0.12}$ | $^{o=6.0}2.29 \pm 0.11$ | 2.89 | 2.92 | 2.43 |
| 0.85 | $^{o=2.2}\mathbf{1.09 \pm 0.15}$ | $^{o=2.0}1.24 \pm 0.15$ | 1.78 | 1.82 | 1.43 |
| 0.80 | $^{o=2.2}\mathbf{0.66 \pm 0.11}$ | $^{o=2.2}\mathbf{0.66 \pm 0.11}$ | 1.05 | 1.08 | 0.86 |
| 0.75 | $^{o=2.2}\mathbf{0.52 \pm 0.03}$ | $^{o=2.2}\mathbf{0.52 \pm 0.03}$ | 0.63 | 0.66 | $\mathbf{0.48 \pm 0.02}$ |
| 0.70 | $^{o=2.2}0.43 \pm 0.07$ | $^{o=2.2}0.43 \pm 0.07$ | 0.42 | 0.43 | $\mathbf{0.32 \pm 0.01}$ |

Table 4: CIFAR10. The number is error percentage on the covered dataset; the lower the better. We see that the superior performance of our method is seen again for another dataset.

| Coverage | Ours (Single Best Model) | Ours (Best per Coverage) | SR | BD | SN |
|---|---|---|---|---|---|
| 1.00 | $^{o=2.0}2.93 \pm 0.17$ | – | 3.58 | 3.58 | 3.58 |
| 0.95 | $^{o=2.0}1.23 \pm 0.12$ | $^{o=1.4}\mathbf{0.88 \pm 0.38}$ | 1.91 | 1.92 | 1.62 |
| 0.90 | $^{o=2.0}\mathbf{0.59 \pm 0.13}$ | $^{o=2.0}\mathbf{0.59 \pm 0.13}$ | 1.10 | 1.10 | 0.93 |
| 0.85 | $^{o=2.0}0.47 \pm 0.10$ | $^{o=1.2}\mathbf{0.24 \pm 0.10}$ | 0.82 | 0.78 | 0.56 |
| 0.80 | $^{o=2.0}\mathbf{0.46 \pm 0.08}$ | $^{o=2.0}\mathbf{0.46 \pm 0.08}$ | 0.68 | 0.55 | $\mathbf{0.35 \pm 0.09}$ |

Table 5: Cats vs. Dogs. The number is error percentage on the covered dataset; the lower the better. This dataset is a binary classification, and the input images have larger resolution.

dataset show misleading performance: on Cats vs. Dogs for $0.8$ coverage, statistical fluctuation caused the validated best model to be one of the worst models on test set.

# 7   Discussion and Conclusion

In this work, we have proposed an end-to-end method to augment the standard supervised classification problem with a rejection option. The proposed method works competitively against the current SOTA [15] but is simpler and more flexible, while outperforming the runner-up SOTA model [14]. We hypothesize that this is because that our model has learned a qualitatively better hidden representation of the data. In Figure 5, we plot the t-SNE plots of a regular model and a model trained with our loss function (more plots in Appendix). We see that, for the baseline, 6 of the clusters of the hidden representation are not easily separable

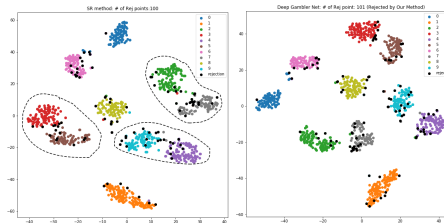

(a) Normal Model    (b) Deep Gambler

Figure 5: t-SNE plot of the second-to-last layer output of a baseline and a deep gambler model for MNIST. Best viewed in color and zoomed-in. The deep gambler model learned a representation that is more separable.

(circled clusters), while a deep gambler model learned a representation with a large margin, which is often associated with superior performance [11, 28, 17].

It seems that there are many possible future directions this work might lead to. One possibility is to use it in scientific fields. For example, neural networks have been used in the classifying neutrinos, and if we do classification on a subset of the data but with higher confidence level, then we can better bound the frequency of neutrino oscillation, which is an important frontier in physics that will help us understand the fundamental laws of the universe [1]. This methods also seems to offer a way to interpret how a deep learning model learns. We can show the top rejected data points at different epochs to study what are the problems that the model finds difficult at different stages of training. Two other areas our method might also turn out to be helpful are robustness against adversarial attacks [37] and learning in the presence of label noise [36, 41]. This work also gives a way incorporate ideas from portfolio theory to deep learning. We hope this work will inspire further research in this direction.

**Acknowledgements:** Liu Ziyin thanks Mr. Zongping Gong for buying him drink sometimes, during the writing of this paper; he also thanks the GSSS scholarship at the University of Tokyo for supporting his graduate study. Z. T. Wang is supported by Global Science Graduate Course (GSGC) program

of the University of Tokyo. This material is based upon work partially supported by the National Science Foundation (Awards #1734868, #1722822) and National Institutes of Health. Any opinions, findings, and conclusions or recommendations expressed in this material are those of the author(s) and do not necessarily reflect the views of National Science Foundation or National Institutes of Health, and no official endorsement should be inferred. Also, This work was supported by KAKENHI Grant No. JP18H01145 and a Grant-in-Aid for Scientific Research on Innovative Areas "Topological Materials Science" (KAKENHI Grant No. JP15H05855) from the Japan Society for the Promotion of Science.

## Footnotes

[1]i.e., we do not consider ensembling methods, but we note that such method can be used together with our method and is likely to increase performance

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
