[Supplementary Material]

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

# Appendix

## 8   A practitioner's guide to our method

### 8.1   Implementation

We give a summary of our proposed method here such that the method can be implemented only reading this section. For a $m$-class classification problem, we propose to change the cross entropy loss to the following modified one:

$$\sum_{i=1}^{m} p_i \log(\hat{p}_i) \rightarrow \sum_{i=1}^{m} p_i \log(\hat{p}_i + \frac{1}{o}\hat{p}_{m+1}) \tag{9}$$

where $\hat{p}_i$ denotes the output prediction of the model for label $i$, and the constraint $\sum_{i=1}^{m+1} \hat{p}_i = 1$ is explicitly enforced by a softmax function over the pre-softmax values; $o$ is a hyperparameter that one can tune, which should be equal to or smaller than $m$ but larger than 1. $\hat{p}_{m+1}$ is the augmented rejection score predicted by the model. For neural networks, this amounts to changing the output dimension from $m$ to $m+1$, and using the loss function above to train the rejection score $p_{m+1}$. All other settings of training may be kept the same. The gambler's loss can be written in Pytorch as the following lines:

```
def gambler_loss(model_output, targets):
    outputs = torch.nn.functional.softmax(model_output, dim=1)
    outputs, reservation = outputs[:,:-1], outputs[:,-1]
    gain = torch.gather(outputs, dim=1, index=targets.unsqueeze(1)) \
                                                        .squeeze()
    doubling_rate = (gain+reservation/reward).log()
    return -doubling_rate.mean()
```

### 8.2   Optimizing the deep gambler's Objective

When training the neural network models, we experimentally found that if $o$ is overly small, the neural network sometimes fails to learn from the training data and converges to a trivial point, which only predicts to abstain, especially on more difficult datasets such as CIFAR10. On CIFAR10, in our grid search experiments, when $o < 6.3$ the trained neural network converged trivially, and when $o \geq 6.3$ the trained model performed at least as well as the ones trained in usual ways. Therefore, in order to converge non-trivially with $o < 6.3$ values, it can be trained with usual cross entropy loss for several epochs in the beginning, and changed to our proposed loss later. This training schedule works well and produces prediction accuracy comparable to large $o$ values. On dataset CIFAR 10, we trained our model with usual cross entropy loss for the first 100 epochs when $o < 6.3$; on dataset SVHN, we trained with cross entropy for the first 50 epochs when $o < 6.0$; on dataset Cats vs Dogs, we trained with cross entropy for the first 50 epochs for all $o$ values.

## 9   Theorem Proofs

**Theorem.** *The optimal doubling rate is given by:*

$$W^*(\mathbf{p}) = \sum p_i \log o_i - H(p) \tag{10}$$

*where $H(p) = -\sum p \log p$ is the entropy of the distribution $p$, and this rate is achieved by proportional gambling $\mathbf{b}^* = p$.*

**Proof**. we have

$$W(\mathbf{b}, \mathbf{p}) = \sum_i p_i \log(b_i o_i)$$

$$= -H(\mathbf{p}) - D(\mathbf{p}\|\mathbf{b}) + \sum_i p_i \log o_i$$

$$\leq \sum_i p_i \log o_i - H(\mathbf{p})$$

and the equality only holds when $\mathbf{b} = \mathbf{p}$.

**Theorem.** *Let $W$ denote the doubling rate given in Def. 2. For a horse race $X$ to which some side information $Y$ is given, the increase $\Delta W$ is:*

$$\Delta W = I(X;Y) = \sum_{x,y} p(x,y) \log \frac{p(x,y)}{p(x)p(y)}. \tag{11}$$

**Proof.** Simply modifying the proof above, it is easy to show that the optimal gambling strategy with side information $Y$ is obtained also by proportional gambling $b(x|y) = p(x|y)$. By definition:

$$W^*(X|Y) = \max_{\mathbf{b}(x|y)} \sum p(x,y) \log(o(x)b(x|y))$$

$$= \sum p(x,y) \log(o(x)p(x|y))$$

$$= -H(X|Y) + \sum p(x) \log o(x)$$

comparing with the doubling rate without side information, we see that the change in doubling rate is:

$$\Delta W = W^*(X|Y) - W^*(X) = H(X) - H(X|Y) = I(X;Y)$$

and we are done.

**Theorem.** *Let $m$ be the number of horses, and let the $W$ be defined as in Eq. 3, and let $o_i = o$ for all $i$, then if $o \geq m$, then the optimal betting always have $b_{m+1} = 0$; if $0 \leq 1$, then the optimal betting always have $b_{m+1} = \|\mathbf{b}\|_1$.*

**Proof.** (1) We first consider the case $o < 1$, we want to show that the optimal solution is to always reserve (i.e. $b_i = 0$ for $i \in \{1, ..., m\}$). Suppose that there exist a solution where $b_i \neq 0$, then the expected return of this part of the bet is then $o_i p_i b_i$, since $p_i \leq 1$, $p_i o_i b_i < 1 = b_i$. This shows that if we instead distribute $b_i$ percentage of our wealth to $b_{m+1}$, then we will achieve better result.

(2) Now we show that for the case $o > m$, we should have $b_{m+1} = 0$. Again, we adopt similar strategy by showing that if there is a solution for which $b_{m+1} \neq 0$, then we can always find a solution better than this. Let $b_{m+1} \neq 0$, and we compare this with a solution in which we distribute $b_{m+1}$ evenly to categories 1 to $m$, the difference in return is:

$$\sum_{i=1}^{m} \frac{p_i b_{m+1} o}{m} - b_{m+1} = \frac{b_{m+1} o}{m} \sum_{i=1}^{m} p_i - b_{m+1} = b_{m+1} \frac{o}{m} - b_{m+1} > 0$$

since $b_{m+1} > m$, and we are done.

## 10    Experiment Detail

For all of our experiments, we use the PyTorch framework[2]. The version is $1.0$. We will release the code of our paper at http://********.

### 10.1    Datasets

**Street View House Numbers (SVHN).** The SVHN dataset is an image classification dataset containing $73,257$ training images and $26,032$ test images divided into 10 classes. The images are digits of house street numbers. Image size is $32 \times 32 \times 3$ pixels. We use the official dataset downloaded by Pytorch utilities.

**CIFAR-10.** The CIFAR10 dataset is an image classification dataset comprising a training set of $50,000$ training images and $10,000$ test images divided into 10 categories. The image size is $32 \times 32 \times 3$ pixels. We use the official dataset downloaded by Pytorch.

**Cats vs. Dogs.** The Cats vs. Dogs dataset is an image binary classification dataset comprising a set of $25,000$ images[3]. As in [15], we randomly choose $20,000$ images as training set and 5000 images as testing set. We resize the size of the images to $64 \times 64 \times 3$.

Figure 6: Error on CIFAR-10 for coverage 0.7 and 0.8 for different $o$. $o$ values below this region require cross entropy loss pretraining to proceed normally, and the results deviate from the above trend lines.

## 10.2 Experiment Setting Details

We follow exactly the experiment setting in [15] and the details are checked against the open source code in [15]. The grid search is done over hyperparameter $o$ with a step size of $0.2$, and the best models of ours for a given coverage are chosen using a validation set. The grid search of $o$ is from 2 to 9.8 on CIFAR-10, from 2 to 8 on SVHN, and from 1.2 to 2 on Cats vs. Dogs. The validation set sizes for SVHN, CIFAR-10 and Cats vs. Dogs are respectively 5000, 2000 and 2000. The standard deviations are estimated by test errors on neighbouring 3 hyperparameter $o$ values in the grid search.[4]

## 11 Additional Experimental Results

### 11.1 Tuning the hyperparameter

The only hyperparameter we introduced is $o$, of which a larger value encourages the model to be less reserved. In this section, we show that $o$ is correlated to the performance of the model at a given coverage and thus is a very meaningful parameter. See figure 6. We see that a lower $o$ causes the model to learn to reject better, but with a larger variance. This suggests that tuning $o$ for different tasks and needs is beneficial. Especially, when $o$ is close to 1, the trained model does not converge to a small training error, and this training error is comparable to its test error. In this case, its resultant total test error rate is increased. However, when $o$ is large, the model does not learn to perform well at low coverages, because the trained abstention score is overly small and affected by numerical error. Therefore, there is a trade-off between total error and error at low coverages, and tuning $o$ is indeed meaningful. Moreover, an appropriate $o$ value encourages the model to learn more from its certain data and learn less from its uncertain data, when compared to the usual cross entropy loss. We believe this is the reason that many of our validated best models outperform the accuracy of the baseline models that use cross entropy loss, even when we train exactly the same models using the same Pytorch package. Therefore, in most situations the best performance is achieved when $o$ is either small or large.

### 11.2 Top-30 Least Certain Images

Here we plot top-30 least certain images in the MNIST training set identified by a trained 4-layer CNN using our method. We also show that how this list changes at epoch $1, 10, 30$. By doing this, we can understand what are the images that the network finds the hardest to identify at different stages of training. We note that the model converges at about 10 epoch.

`https://github.com/Z-T-WANG/NIPS2019DeepGamblers`

Figure 7: Epoch 1. The top-30 images in the MNIST testing set found by Deep Gamblers, with the uncertainty score at the top.

Figure 8: Epoch 10. The top-30 images in the MNIST testing set found by Deep Gamblers, with the uncertainty score at the top.

Figure 9: Epoch 30. The top-30 images in the MNIST testing set found by Deep Gamblers, with the uncertainty score at the top.

(a) Normal Model                                    (b) Deep Gambler

Figure 10: t-SNE plot of a normal model and the same model trained using our loss function.

(a) $h$ set to reject 30 points                     (b) $h$ set to reject 100 points

Figure 11: t-SNE plot of a regular deep model with different $h$.

## 11.3 t-SNE Plots

The experiment is done on MNIST with a 4-layer CNN model trained until convergence (10 epochs). The t-SNE (we used sklearn and its default parameters) plot is based on the output of the second-to-last layer. The raw t-SNE of the two models are in Figure 10. The points rejected by SR on the simple model is given in Figure 11 and that by a deep gambler model is in Figure 12. We see the the normal model has not learned a representation that is easily separable (circled clusters), while a deep gambler model learned a representation with a large margin, which is often associated with superior performance [11, 28, 17]. From the plot, we notice that the baseline model seems to have mixed up 4 with 9, 3 with 5, and 2 with 7. More interestingly, we also note that we can also use the SR method on a deep gambler model, and we notice that in this case the rejected points are almost always the same. Suggesting that the reason for our superior performance is due to learning a better representation.

(a) $h$ set to reject 30 points          (b) $h$ set to reject 100 points

Figure 12: t-SNE plot of a deep gambler model with different $h$.