[Reviews · NeurIPS 2019]

Reviewer 1



The approach presented in this paper is interesting and it seems that the portfolio selection approach naturally fits into the selective classification network. The paper is well motivated, well-written, clear, and self-contained. The reason I chose this score (marginally below the accept threshold) is because of the following two reasons: 1.The approach you suggested (adding a new class) is very similar to the selectivenet approach, and as we can observe in the experimental part, your approach leads only to marginal improvement. 2. Your approach is indeed simpler than selective net. However, the table on page 2 is a bit misleading. The gain of selective net, as explained in their paper, is due of the fact that for a given coverage the net needs to be retrained. Summarizing the above, the contribution over the SOTA (selectivenet) is marginal.

Reviewer 2



This paper tackles an important problem; methods aiming to improve the models' ability to express uncertainty are of high importance to the field. The proposed method is simple, flexible and seemingly novel. The method does not cover/evaluate the regression problems which might limit its impact scope. Even though the obtained results are not outperforming the state-of-the-art, this line of research on incorporating the portfolio theory to the confidence estimation is interesting and can inspire future studies. The paper is sometimes difficult to follow and presentation quality of the paper and its clarity can be improved.

Reviewer 3



Originality: this is in my opinion a quite strong point of the paper, that nicely bridges two different theories and brings an interesting interpretation to rejection samples and their nature within the learning theory. Quality is good, even if some typos remain here and there (L114,L156,L200,L272) Clarity: the authors often speak of assessing the uncertainty associated to a sample, however what they provide is a scalar-evaluation allowing to assess an overall confidence we can attach to a given instance being properly classified (and then possibly rejecting it according to our policy). We remain far from a full assessment of the uncertainty as would give a calibrated probability distribution or a gradual conformal prediction (that would go from the null set to the totally empty one). It could be useful if authors were clearer about that from the start. Significance: from the paper, I had the feeling that theorems 1 and 2 are quite direct trasnposition of known results, while theorem 3 is kind of obvious (if every horse give me more than m times my money, then for sure a uniform bet on them will make me win without needing any saving, and if I cannot win more than one by betting on any of them, it is of course better to keep my money). So it could be argued that those main results are in fact expected. Would authors agree with that? This said, I think the bridges that are made are sufficient in significance by thmeselves.

[Author Response · NeurIPS 2019]

General Comment: We thank all the reviewers for providing comments that have been helpful for us to reassess the strengths and weaknesses of the DeepGambler method and writing. The most important message of this paper is that various connections between assessing prediction uncertainty in deep learning and ideas from portfolio theory can be drawn naturally, and the contribution lies in the connections and experiments.

**Better representations when compared with SelectiveNet [R1].**
Our proposed DeepGambler model learns representations that are very different from SelectiveNet. This is exemplified in Figure 1 where on the left we show the representations from DeepGambler (rejected points in black) and on the right we show the representations from SelectiveNet (rejected points shown in color, taken from SN paper). It is very interesting to see that our proposed DeepGambler model preserves the semantic differences of the rejected points: the rejected points are still in close to their respective clusters, and are not attracted to each other. In sharp comparison, the SelectiveNet method seems to discard relevant information about the variations of the data points by clustering all the rejected points together. This is an important difference between our DeepGambler approach and previous work: qualitatively speaking, we argue that our method

(b) Deep Gambler                    (a) SelectiveNet

Figure 1: t-SNE plot of the penultimate layer representation

learned better representation of the rejected points. This can explain why the DeepGambler model seems to have slightly better full coverage performance (see paper, tables 4 and 5, first row).

**State-of-the-art performance where it matters the most [R1, R2, R3].** Our proposed approach is outperforming prior approaches (including previous SOTA Selective Net) in a statistically significant way for all datasets, for the most critical categories of 90% and 95%. While we may have understated these important results in the paper, we believe these categories (90% and 95%) are the most critical for real-world applications: modern applications often involve a very large number of datapoints (e.g., 1+ million), and it would be hard to imagine more than 10% of the data points being passed to a human expert (or a more expensive model). The performance of our proposed approach is still very competitive for the lower categories with an overall performance (over 14 categories): Our proposed DeepGambler approach is better (statistically significant) in 9, comparable in 4 and is outperformed in 1 only one case (for a coverage of 70%). **Simpler yet strong single model [R1].** One practical advantage of DeepGambler over SN is that a single model can be used for various coverages. We point out that this simplicity does not compromise the performance of the model. In fact, a single DeepGambler model, trained once, can outperform SN trained for different coverages. Compare column 1 and 5 in table 3, 4, 5, we also see that DeepGambler dominates SN in most categories.

**Comparison with other methods in Figure 3 [R2].** In fact, both figure 3 and 4 are for demonstrating how our model works, not for bench-marking against other models. That said, some qualitative comparison are available. This experiment shows how DeepGambler behaves compared with the SR method. One can show that the ES behaves exactly the same as SR in binary classification, and therefore the figure 3 reflects how SR would perform in this toy task. Also, we gave more comment on the similarity and difference between the SR and the PM in section 11.3 in the appendix; in fact, this experiment shows that, learning a hidden representation to predict an uncertainty score is better than simply calculating a score from the raw prediction. **For Figure 4 [R2].** This is also a functional demonstration of the DeepGambler, and, in fact, we used this as a sanity check to check whether our method is doing what it should. This can be directly compared with Figure 4 in the BD paper (notice that in the BD paper, the experiment is also purely for demonstration). Qualitatively speaking, it looks like the PM behaves in a similar way to BD in this task, and it would be quite surprising (and, in a bad way) to imagine if other method such as SN would behave in any different way. Please also refer to the 2rd paragraph for a qualitative comparison between DeepGambler and SN.

**The effect of changing $o$ [R2].** Indeed, lower values of o show better results, as is shown in figure 6 in the appendix, where we conducted grid search over $o$ on CIFAR 10. The linear fit shows a clear drop in testing loss when using smaller $o$. However, from the same figure one see that, while the averaged loss drops steadily when $o$ becomes small, its variance (with respect to different random seed) increases quickly as $o$ drops. We hypothesize that there is some implicit bias-variance tradeoff in our proposed method, and similar grid search results are also observed in the other two datasets. Therefore, due to larger variance at lower $o$, sometimes models with larger $o$ are chosen by the validation process. We expect models with lower $o$ to be chosen had we conducted sufficient grid search over the random seed (we only performed minimal grid search currently).

**Meaning of Uncertainty [R3].** Yes, it would have been better if we were clearer about the meaning of the "uncertainty" from the beginning, it indeed refers to predicting a confidence score instead of a statistical uncertainty, and so it is meaningful when compared to the confidence scores of another data points. We will use "confidence score" when revising the manuscript.

[Meta-Review · NeurIPS 2019]

Reviewers and the area chair agree that this paper is strong. A significant strength is that the method is simple enough that practitioners can try using it immediately. The connection between ML and portfolio theory will be enlightening to researchers who are unaware of it.